# The effect of systematic antidepressant treatments in the early stages on sleep and impulsivity in bipolar euthymic patients: A cross-sectional study

Mingjin Wang[1,2‡], Xuguang An[1‡], Dongyu Han[1], Xiaofei Hou[1], Chenghao Yang[1,2]*

**1** Tianjin Anding Hospital, Mental Health Center of Tianjin Medical University, Tianjin, China, **2** Tianjin Medical University, Tianjin, China

‡ The two authors share the first authorship.
* yts83420@163.com

## Abstract

### Background and Objectives

Due to early misdiagnosis, bipolar patients who had a depressive episode as their initial onset often received systematic antidepressant treatments and continued to suffer from sleep disturbances and elevated impulsivity, even during euthymic state. The study aims to assess the effect of systematic antidepressant treatments in the early stages on sleep and impulsivity in bipolar euthymic patients, and further explore the potential mediating role of sleep in the relationship between early antidepressant uses and impulsivity.

### Methods

A total of 124 bipolar euthymic patients were enrolled. Based on the early use of antidepressants, patients were divided into AT group (systematic antidepressant treatment group) and NT group (no systematic antidepressant treatment group). Sleep quality and impulsivity were assessed using Pittsburgh Sleep Quality Index and Barratt Impulsivity Scale Questionnaire version 11, respectively. Statistical analyses were conducted using the t-test, Chi-square test, and Mann-Whitney U test, and mediation analysis was performed using bootstrapping.

### Results

Patients in the AT group showed poorer sleep quality and higher impulsivity than those in the NT group. Patients' sleep quality was positively correlated with impulsivity. Sleep quality mediated the relationship between antidepressant uses and impulsivity, including both overall impulsivity and non-planning impulsivity.

**Data availability statement:** All relevant data are within the paper and its Supporting Information files.

**Funding:** The work was supported by program of "Research Plan Project of Tianjin Municipal Education Commission" (2022KJ264). The funders had no role in study design, data collection and analysis, decision to publish, or preparation of the manuscript.

**Competing interests:** The authors have declared that no competing interests exist.

## Conclusions

This study suggests a correlation between early-stage antidepressant use, sleep quality, and impulsivity of bipolar euthymic patients, highlighting the importance of early diagnosis of bipolar disorder and appropriate antidepressant prescriptions. Furthermore, improving sleep quality would be effective in reducing the risk of impulsive behaviors.

## 1. Introduction

Bipolar disorder (BD) is a life-long chronic and disabling disease characterized by recurring, shifting episodes of depression, mania, or hypomania, affecting over 1% of the global population [1]. It is often associated with impaired function and neuro-cognition [1,2], partially due to dysregulation in sleep and impulsivity [3–5]. According to reports, from the first episode to the diagnosis of BD, there is a delay about 5–10 years or more [6–8]. In clinical practice, approximately 50% of BD patients initially present with a depressive episode and more than 20% are misdiagnosed as major depressive disorder (MDD) [9–11]. Therefore, due to the incorrect or delayed diagnosis, many patients receive routine systematic treatments with antidepressants during the early stages [10,12]. As is well known, the use of antidepressants in treating bipolar depression can eventually lead to several negative outcomes, including accelerated mood switching, irritability, cognitive impairment, and increased suicidal risk [7,13,14]. Notably, the potential effects of antidepressant use in the early phase have not received much attention.

Sleep disturbances commonly occur in individuals with BD and can be present throughout the entire course of the disease [15–17]. Poor sleep quality is comprehensively correlated with depressive, hypomanic, and manic episodes, and changes in sleep patterns are common symptoms [15,18]. Sleep disturbances have been demonstrated to be a significant factor contributing to the low quality of life [16,19,20]. For instance, individuals with poor sleep quality showed deficits in emotion regulation, less positive affect, and increased impulsivity [19,21]. Additionally, as a trait of BD, impulsivity is observed throughout the entire disease course [22,23], resulting in a notable impact on cognitive function and suicide attempts [24,25]. The link between sleep disturbances and impulsivity in bipolar patients has been previously reported; individuals with poor sleep quality are more likely to be impulsive [26–29]. Such correlations are comprehensive and observed across multiple disease cohorts. One study focusing on inpatients with antisocial or borderline personality disorder indicated that sleep problems were significantly related to self-reported impulsivity [26]; and improved sleep duration was associated with statistical reductions in aggressive thoughts and behaviors in adolescents with substance dependence [30]. However, extant research examining the association between sleep quality and impulsivity during the euthymic phase of BD is sparse.

Overall, early diagnosis of BD presents significant challenges, leading to widespread use of antidepressant medications. It is of considerable clinical value to

explicitly identify their potential adverse effects. Furthermore, the prevalence of sleep disturbances and increased impulsivity, as well as their effect on social functioning, should receive due attention. Additionally, based on the principles of comprehensive care management, growing interest has focused on abnormalities implicated in the euthymic state of bipolar patients. In this regard, this study aimed to explore the influence of systematic treatment with antidepressants (defined as receiving adequate dosage of antidepressants for at least 6 weeks) in the early stages (defined as a period from initial depressive onset to the diagnosis of BD) on sleep and impulsivity, and to investigate the potential relationship between sleep disturbances and dysregulated impulsivity of bipolar euthymic patients. The goal is to provide guidance for a deeper understanding of antidepressant use in BD patients, as well as for the maintenance treatment of bipolar euthymic patients.

## 2. Materials and methods

### 2.1. Setting and subjects

The study consists of 124 bipolar euthymic patients who had experienced a depressive episode as their initial onset. Based on a history of systematic antidepressant treatments in the early stages, patients were consecutively divided into the systematic antidepressant treatment (AT) group and the no systematic antidepressant (NT) group. All patients were aged 18–50, recruited from outpatient and inpatient departments of Tianjin Anding Hospital located in Tianjin, China, during 1st September, 2020 and 30th May, 2023. After approving by the Medical Ethics Committee of Tianjin Anding Hospital, written informed consent from all patients was obtained.

The patients enrolled in this study had already been diagnosed with BD according to the Diagnostic and Statistical Manual of Mental Disorders, Fourth Edition (DSM-IV-TR) with Structured Clinical Interview for DSM-IV (SCID). They had been in remission for at least 4 weeks prior to engage in the study. We used the Mini-International Neuropsychiatric Interview (MINI) to retrospectively diagnose the initial depressive episode. All patients were assessed with the Montgomery-Asberg Depression Scale (MADRS) and the Young Mania Rating Scale (YMRS) to determine their euthymic status, with both scores being lower than 7 points [31,32]. The Hypomania Check List-32 items (HCL-32) was administered to screen for potential hypomanic symptoms during the early stages. Patients who had organic mental disorders, mental retardation, personality disorders, cognition impairments, concurrent or a history of substance abuse or dependence, or comorbidity of attention deficit and hyperactivity disorder (ADHD), were excluded. Please refer to the published study protocol for experimental details [33].

### 2.2. Procedures and clinical assessment

This is a cross-sectional study aimed at exploring the influence of systematic treatment with antidepressants in the early stages on sleep and impulsivity, as well as the mediating role of sleep quality in the relationship between early-stage antidepressant use and impulsivity in bipolar euthymic patients. Two senior psychiatrists initially screened patients according to the inclusion and exclusion criteria. Afterward, all enrolled patients signed informed consent and were divided into the AT and NT groups. In addition, the senior psychiatrists collected other study parameters, including age, sex, ethnicity, age of onset and diagnosis, and a history of mood stabilizer use. The remaining scales were then administered by other investigators, who were blinded to the group assignments. The Pittsburgh Sleep Quality Index (PSQI), a self-rating questionnaire with 18 items, was used to assess sleep disturbances and quality over the past month [34]. Barratt Impulsivity Scale Questionnaire version 11 (BIS-11A) is a self-reported scale used to assess impulsivity [35], consists of 30 questions with a 4-point scale for each response, ranging from 1 to 4. It includes three subscales: attention impulsivity, motor impulsivity, and non-planning impulsivity (**Fig 1**).

### 2.3. Statistical analysis

The baseline characteristics of the two groups were compared. The normality of the data was tested using the Shapiro-Wilks test. If the sample was normally distributed, the data was expressed as "mean ± standard deviation", and the t-test

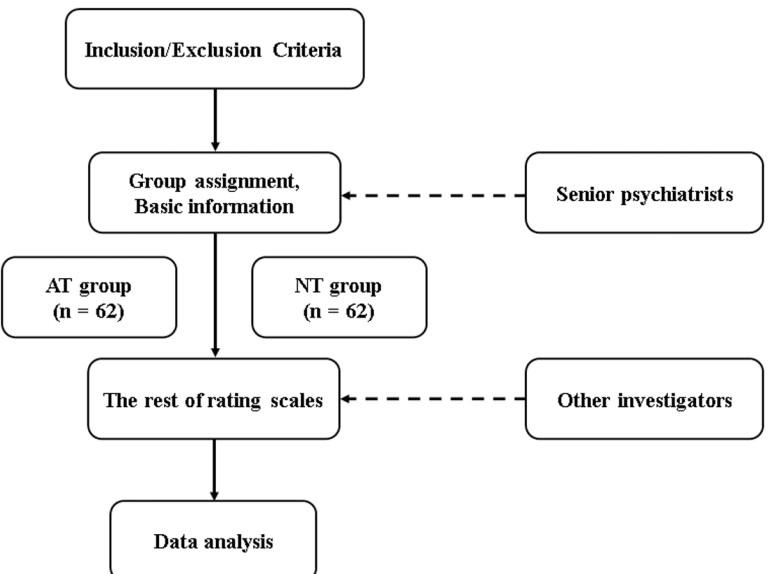

**Fig 1. Study flowchart.** AT, systematic antidepressant treatment; NT, no systematic antidepressant treatment.

was used to compare continuous variables. The Pearson correlation analysis was performed to examine the bivariate correlations between the variables. If the sample was not normally distributed, the data was expressed as the median (inter-quartile range). The difference between the two groups was compared using the Mann-Whitney U test, and the bivariate correlations between the variables were tested using Spearman correlation analysis. For categorical variables, the Chi-square test was used to analyze the differences between groups.

To explore potential mediating effects, we utilized a bootstrapping technique with Model 4 of the PROCESS macro, version 4.1. To robustly assessing mediation, we generated 5000 bootstrap estimates alongside bias-corrected 95% confidence intervals (CIs). Non-zero CI values indicated a significant mediation effect at the 5% significance level. To control for confounding factors, we adjusted for covariates such as sex, age, years of diagnostic delay, and the use of mood stabilizers in the mediation analysis. All p-values were two-tailed, with statistical significance set at $p<0.05$. The analyses were conducted using SPSS, version 26.0.

## 3. Results

### 3.1. Descriptive statistics

The sociodemographic and clinical characteristics of all 124 participants are presented in **Table 1**. There was no statistically significant difference between the two groups ($p>0.05$) in terms of age, sex, years of diagnostic delay, use of mood stabilizers, and scores of MADRS and YMRS. The average PSQI score of the AT group (Mean=8.27) was higher than the NT group (Mean=6.13), and the difference was statistically significant ($p<0.001$). The participants in the AT group had significantly higher level of impulsivity than those in the NT group. The average impulsivity score was 61.39 (SE=9.429) for the AT group and 57.19 (SE=8.596) for the NT group. The average subscale scores for the AT group were 13.29 (SE=2.939) for attention impulsivity, 23.13 (SE=3.957) for motor impulsivity, and 24.97 (SE=4.949) for non-planning impulsivity. The average subscale scores for the NT group were 12.11 (SE=2.776) for attention impulsivity, 22.39 (SE=3.800) for motor impulsivity, and 22.69 (SE=4.779) for non-planning impulsivity. The scores of attention impulsivity, non-planning impulsivity, and overall impulsivity measured by BIS-11A were significantly different between the two groups ($p<0.05$), with

**Table 1. Sociodemographic and clinical characteristics.**

| Variables | NT (n = 62) | AT (n = 62) | t/Z/χ² | p |
|---|---|---|---|---|
| Age | 30.00 (23.75-37.00) | 28.50 (22.00-35.25) | -0.713 | 0.476 |
| Sex (women/men) | 38/24 | 35/27 | 0.300 | 0.584 |
| Diagnostic delay (years) | 3.00 (1.00-6.00) | 3.00 (1.00-7.00) | -0.314 | 0.753 |
| Use of mood stabilizer(yes/no) | 9/53 | 11/51 | 0.238 | 0.625 |
| Depressive symptoms (MADRS) | 4.00 (1.00-7.00) | 3.00 (0.00-6.00) | -0.930 | 0.352 |
| Manic symptoms (YMRS) | 4.00 (3.00-5.00) | 4.00 (2.00-5.00) | -0.931 | 0.352 |

Note: MADRS, Montgomery-Asberg Depression Rating Scale; YMRS, Young Mania Rating Scale.

individuals in the AT group being significantly more impulsive. However, no statistical difference was found in motor impulsivity between the two groups ($p>0.05$). See **Table 2** for details.

### 3.2. Preliminary correlation analyses

As shown in **Table 3**, systematic treatment with antidepressants was positively correlated with the total score of the PSQI (r=0.488, $p<0.000$), indicating that antidepressant use in the early stages led to worse sleeping quality in bipolar euthymic patients. Furthermore, with the exception of motor impulsivity (r=0.128, $p>0.05$), a positive correlation was also observed between systematic antidepressant treatment and the scores of BIS-11A (r=0.245, $p<0.01$), attention impulsivity (r=0.190, $p<0.05$) and non-planning impulsivity (r=0.228, $p<0.05$). In other words, antidepressant use in the early stages can significantly elevate the level of impulsivity in bipolar euthymic patients. Finally, sleeping quality, as measured by PSQI, was positively correlated with the strength of impulsivity (r=0.290, $p<0.001$) and each subtype, including attention impulsivity (r=0.191, $p<0.05$), non-planning impulsivity (r=0.289, $p<0.001$), and motor impulsivity (r=0.186, $p< 0.05$).

### 3.3. Mediation analyses

The results of the mediation analyses are displayed in **Table 4**, **Fig 2** and **Fig 3**. Model 1 shows the association between independent variable (systematic antidepressants treatment) and the mediator (sleep quality by PSQI). Model 2 shows the relationship between the independent variable and the dependent variables (scores of BIS-11A, attention impulsivity, and non-planning impulsivity). Model 3 shows the correlation between the independent variable, the mediator and the dependent variables. Both crude and adjusted models are presented. As shown in the crude model, there is a significant association between antidepressant use and PSQI (β=2.145, SE=0.438, $p<0.000$). In Model 1, after adjusting for sex, age, years of diagnostic delay and use of mood stabilizers, antidepressant use remained significantly associated with PSQI (β=2.177, SE=0.446, $p<0.000$). Similarly, the associations between antidepressant use and the scores of BIS-11A (β=4.194, SE=1.620, $p<0.05$), attention impulsivity (β=1.177, SE=0.513, $p<0.05$), and non-planning impulsivity (β=2.274, SE=0.874, $p<0.05$) were significant in the crude model. In Model 2, these significant associations remained after adjusting for sex, age, years of diagnostic delay and use of mood stabilizers (BIS-11A: β=4.357, SE=1.613, $p<0.05$; attention impulsivity: β=1.234, SE=0.516, $p<0.05$; non-planning impulsivity: β=2.316, SE=0.871, $p<0.05$). In Model 3, after correcting for sex, age, years of diagnostic delay, and use of mood stabilizers, no significant correlations were found between antidepressant use and the scores of BIS-11A (β=2.701, SE=1.736, $p>0.05$), attention impulsivity (β=0.971, SE=0.565, $p>0.05$), or non-planning impulsivity (β=1.426, SE=0.938, $p>0.05$). The positive correlation between PSQI and attention impulsivity remained statistically insignificant (β=0.121, SE=0.106, $p>0.05$), while the associations with BIS-11A (β=0.761, SE=0.327, $p<0.05$) and non-planning impulsivity (β=0.409, SE=0.177, $p<0.05$) remained statistically significant.

The total, direct, and indirect effects of the mediation analyses are displayed in **Table 5**. In the crude model, both the indirect effects of overall impulsivity and non-planning impulsivity were significant because the bootstrap 95% CIs did not

**Table 2. Sleeping quality and impulsivity.**

| Variables | NT (n = 62) | AT (n = 62) | t/Z | p |
|---|---|---|---|---|
| PSQI | 6.00 (5.00-7.00) | 9.00 (7.00-10.00) | -5.414 | 0.000 |
| BIS-11A | | | | |
| Total score | 57.19±8.596 | 61.39±9.429 | -2.588 | 0.011 |
| Motor impulsiveness | 21.50 (20.00-24.25) | 22.50 (21.00-25.25) | -1.417 | 0.156 |
| Attention impulsiveness | 12.00 (10.00-14.00) | 13.00 (11.00-15.25) | -2.106 | 0.035 |
| Non-planning impulsiveness | 22.69±4.779 | 24.97±4.949 | -2.603 | 0.010 |

Note: PSQI, Pittsburgh Sleep Quality Index; BIS-11A, Barratt Impulsiveness Scale 11-A.

**Table 3. Correlations between key study variables.**

| | 1 | 2 | 3 | 4 | 5 | 6 |
|---|---|---|---|---|---|---|
| 1. Antidepressant uses | 1 | | | | | |
| 2. PSQI | 0.488** | 1 | | | | |
| 3. BIS-11A total score | 0.245** | 0.290** | 1 | | | |
| 4. Motor impulsivity | 0.128 | 0.186* | 0.729** | 1 | | |
| 5. Attention impulsivity | 0.190* | 0.191* | 0.689** | 0.307** | 1 | |
| 6. Non-planning impulsivity | 0.228* | 0.289* | 0.856** | 0.409** | 0.447** | 1 |

Note: * $p < 0.05$ (two-tailed test); ** $p < 0.01$ (two-tailed test); PSQI, Pittsburgh Sleep Quality Index; BIS-11A, Barratt Impulsiveness Scale 11-A.

**Table 4. Mediating effect of sleep between antidepressant uses and impulsivity.**

| | β | SE | t | p | LLCI | ULCI | p-adjusted |
|---|---|---|---|---|---|---|---|
| **Model 1: X→M** | | | | | | | |
| Antidepressant uses → Sleep | 2.145 | 0.438 | 4.899 | 0.000 | 1.278 | 3.012 | 0.000 |
| **Model 2: X→Y** | | | | | | | |
| Antidepressants uses → BIS-11A total score | 4.194 | 1.620 | 2.588 | 0.011 | 0.986 | 7.401 | 0.008 |
| Antidepressants uses → Attention impulsivity | 1.177 | 0.513 | 2.293 | 0.024 | 0.161 | 2.194 | 0.018 |
| Antidepressants uses → Non-planning impulsivity | 2.274 | 0.874 | 2.603 | 0.010 | 0.545 | 4.004 | 0.009 |
| **Model 3: X, M→Y** | | | | | | | |
| Antidepressants uses → BIS-11A total score | 2.541 | 1.741 | 1.459 | 0.147 | -0.906 | 5.987 | 0.122 |
| Antidepressants uses → Attention impulsivity | 0.899 | 0.561 | 1.604 | 0.111 | -0.211 | 2.009 | 0.088 |
| Antidepressants uses → Non-planning impulsivity | 1.384 | 0.939 | 1.474 | 0.143 | -0.475 | 3.242 | 0.131 |
| Sleep → BIS-11A total score | 0.771 | 0.329 | 2.342 | 0.021 | 0.119 | 1.422 | 0.022 |
| Sleep → Attention impulsivity | 0.130 | 0.106 | 1.226 | 0.223 | -0.080 | 0.340 | 0.259 |
| Sleep → Non-planning impulsivity | 0.415 | 0.178 | 2.339 | 0.021 | 0.064 | 0.766 | 0.022 |

Note: BIS-11A, Barratt Impulsiveness Scale 11-A; p-adjusted, p value of adjusting age, sex, years of diagnostic delay and use of mood stabilizers.

contain zero, which suggests that the mediation models were valid. The significance remained even after adjusting for covariates, including sex, age, years of diagnostic delay, and use of mood stabilizers. The mediation analyses indicated that systematic antidepressant treatment in the early stages could increase impulsivity by affecting sleep quality in bipolar euthymic patients.

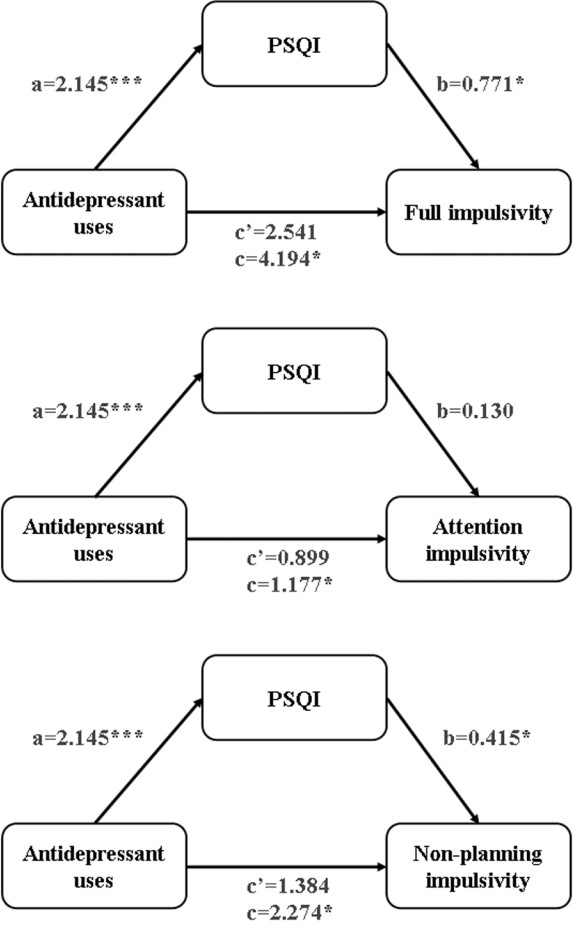

**Fig 2. The crude mediation model of the association of antidepressant uses and impulsivity through sleep quality.** PSQI, Pittsburgh Sleep Quality Index; $p^* < 0.05$, $p^{***} < 0.001$ (two-tailed).

## 4. Discussion

The current study explored the effects of systematic antidepressant treatment in the early stages on sleeping quality and impulsivity, as well as the mediation role of sleeping quality in the relationship between antidepressant use and impulsivity in patients with BD during the euthymic phase. The main findings of this study are as follows: First, participants' sleep quality was significantly positively correlated with impulsivity; second, systematic antidepressant treatment in the early stages had a noticeably negative influence on sleep and both overall impulsivity and its subtypes; third, the effect of antidepressant use on overall impulsivity and non-planning impulsivity was mediated by sleep quality. To our knowledge, this is the first study to date investigating the correlations between antidepressant use, sleep quality, and impulsivity in bipolar euthymic patients.

Impulsivity, defined as a predisposition toward unplanned reactions to internal or external stimuli and a lack of regard for negative consequences [36], often leads to serious behavioral outcomes, such as violent actions and suicide attempts [37,38]. Impulsivity is observed throughout the course of BD, including during the euthymic state [39,40]. The link between sleep and impulsivity is well-documented, yet studies focusing on both factors in remitted BD patients are sparse. Research indicates that bipolar euthymic patients with poor sleep quality exhibit higher impulsivity, similar

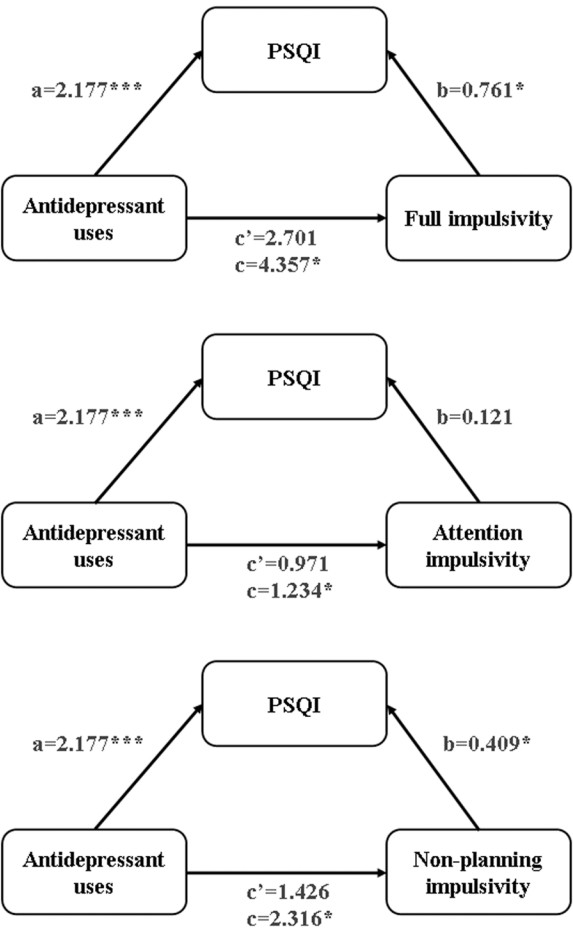

**Fig 3. The adjusted mediation model of the association of antidepressant uses and impulsivity through sleep quality.** PSQI, Pittsburgh Sleep Quality Index; adjusting sex, age, years of diagnostic delay and use of mood stabilizers; *p* * < 0.05, *p* *** < 0.001 (two-tailed).

to findings by Anda Gershon et al., who showed increased impulsivity in adolescents with BD due to variation in sleep duration [25]. A study by Laura Palagini et al. in bipolar patients with mixed features reaches a unanimous conclusion [41]. Additionally, research across various demographics, including healthy individuals, as well as those with ADHD and substance abuse, shows a significant correlation between sleep disruption and increased impulsivity [3,29,30,42]. Sleep disturbances impact prefrontal cortical functions that are crucial for regulating behavior and emotions [26,43,44], which explains heightened impulsivity in those with poor sleep. Moreover, studies suggest a bi-directional relationship between sleep quality and impulsivity [45–47], offering valuable insights, even though these findings are not directly generalizable from healthy individuals to BD patients. A positive correlation was observed between MADRS and PSQI scores (r=0.307, p < 0.001), excluding MADRS from adjusted confounder analysis. Research suggests a strong bi-directional link between sleep and depression; insomnia is not only a symptom but also a risk factor for depression, and conversely, depression commonly causes insomnia [48,49]. Additionally, Nierenberg et al. reported that sleep issues are prevalent residual symptoms in depressives achieving clinical remission (scores of the Hamilton Depression Rating Scale ≤ 7) [50]. This reciprocal relationship may explain why more severe depressive symptoms correlate with poorer sleep quality during bipolar euthymia.

**Table 5.  The results of the mediation analyses.**

|  | Effect | SE | LLCI | ULCI | LLCI-adjusted | ULCI-adjusted |
|---|---|---|---|---|---|---|
| **Antidepressants on BIS-11A** |  |  |  |  |  |  |
| Total effect | 4.194 | 1.620 | 0.986 | 7.401 | 1.164 | 7.551 |
| Direct effect | 2.541 | 1.741 | -0.906 | 5.987 | -0.736 | 6.139 |
| Indirect effect | 1.653 | 0.955 | 0.187 | 3.880 | 0.099 | 3.882 |
| **Antidepressants on Attention impulsivity** |  |  |  |  |  |  |
| Total effect | 1.177 | 0.513 | 0.161 | 2.194 | 0.212 | 2.255 |
| Direct effect | 0.899 | 0.561 | -0.211 | 2.009 | -0.148 | 2.090 |
| Indirect effect | 0.279 | 0.282 | -0.191 | 0.904 | -0.235 | 0.937 |
| **Antidepressants on Non-planning impulsivity** |  |  |  |  |  |  |
| Total effect | 2.274 | 0.874 | 0.545 | 4.004 | 0.592 | 4.041 |
| Direct effect | 1.384 | 0.939 | -0.475 | 3.242 | -0.431 | 3.283 |
| Indirect effect | 0.890 | 0.560 | 0.076 | 2.238 | 0.087 | 2.248 |

Note: BIS-11A, Barratt Impulsiveness Scale 11-A; LLCI-adjusted, LLCI of adjusting age, sex, years of diagnostic delay and use of mood stabilizers; ULCI-adjusted, ULCI of adjusting age, sex, years of diagnostic delay and use of mood stabilizers.

The use of antidepressants in bipolar depression presents a paradox. About 20%-50% of BD patients intermittently or chronically exhibit subsyndromal symptoms, including subthreshold depression, cognitive impairment, and circadian rhythm disruptions [51–53]. Evidence indicates that antidepressants may increase aggression and impulsivity [54–57], although other studies associate SSRIs with reduced impulsivity [57–59]. It seems that antidepressants can both disrupt and promote sleep [14]. This may be disease status-dependent and also associated with the types and pharmacological mechanisms of the drugs. The comprehensive therapeutic effect of antidepressants is to alleviate depressive symptoms, including both positive and negative affect, which are significantly associated with impulsivity. However, antidepressant medications themselves can induce aggressive symptoms. The current study focused on sleeping quality and impulsivity during the euthymic period, which may be influenced by the antidepressant use in the early stages.

The current study provides the first demonstration that sleep plays a mediating role in the effects of antidepressant use on impulsivity in bipolar euthymic patients. Additionally, compared to previous studies, we specifically describe the relationship between sleep quality and three subtypes of impulsivity. While our results provide valuable insights, there are some limitations that need to be considered. First, the lack of objective measurements of assessing sleep problem and impulsivity, as well as the reliance on self-reported data, may introduce potential bias. Second, the cross-sectional nature of our study design limits our ability to establish causation, restricting our ability to elaborate on temporal relationships between antidepressant use, sleep, and impulsivity. Lastly, other factors may interfere with the study outcomes but were not considered, such as discrepancies in antidepressant types, dosages, and the duration of antidepressant treatment. Future research should incorporate longitudinal observations, utilize objective indicators, and minimize confounding factors to enhance the study's validity.

Overall, the difficulty in diagnosing BD early and the inconsistent attitude toward the administration of antidepressants to bipolar depressive patients pose intense challenge, which has resulted in an extensive use of antidepressants among individuals with bipolar depression. The conclusions drawn from our research underscore the importance of caution when considering the use of antidepressants in BD patients, even before the manifestation of distinctive symptoms of BD. Furthermore, sleep quality can be targeted as an intervention to mitigate the potential effects of early antidepressant treatment on impulsivity, thereby serving as a compensatory mechanism to reduce impulsive behaviors. The study findings will be beneficial in clinical practice for future applications.

## Supporting information

**S1 File. Supplementary data.**
(XLSX)

## Acknowledgments

We would like to express our gratitude to all the participants for their support and dedication to this study.

## Author contributions

**Conceptualization:** Chenghao Yang.

**Data curation:** Mingjin Wang, Xuguang An.

**Formal analysis:** Dongyu Han, Xiaofei Hou.

**Funding acquisition:** Chenghao Yang.

**Methodology:** Dongyu Han, Xiaofei Hou.

**Supervision:** Chenghao Yang.

**Writing – original draft:** Mingjin Wang, Xuguang An.

**Writing – review & editing:** Chenghao Yang.

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
