## [Decision Letter · Decision Letter 0]

5 Feb 2025

PONE-D-24-50851The impact of systematic antidepressant treatments in early stage on sleep and impulsivity in bipolar euthymic patients: A cross-sectional studyPLOS ONE

Dear Dr. Yang,

Thank you for submitting your manuscript to PLOS ONE. After careful consideration, we feel that it has merit but does not fully meet PLOS ONE’s publication criteria as it currently stands. Therefore, we invite you to submit a revised version of the manuscript that addresses the points raised during the review process.

I kindly request that you carefully review the English language and make any necessary corrections. In addition, please revise the manuscript according to the comments and suggestions provided by the reviewers. I also encourage you to consider incorporating more recent references where appropriate to ensure your work reflects the current state of the field. Furthermore, I would appreciate it if you could expand upon the future directions section, providing more detailed insights into potential avenues for future research stemming from your findings.

We look forward to receiving your revised manuscript.

Kind regards,

Giuseppe Marano

Academic Editor

PLOS ONE

Reviewers' comments:

Reviewer's Responses to Questions

**Comments to the Author**

1. Is the manuscript technically sound, and do the data support the conclusions?

Reviewer #1: Partly

Reviewer #2: Yes

2. Has the statistical analysis been performed appropriately and rigorously? 

Reviewer #1: Yes

Reviewer #2: Yes

3. Have the authors made all data underlying the findings in their manuscript fully available?

Reviewer #1: Yes

Reviewer #2: Yes

4. Is the manuscript presented in an intelligible fashion and written in standard English?

Reviewer #1: No

Reviewer #2: Yes

5. Review Comments to the Author

Reviewer #1: First of all, I would like to thank Authours for providing this best topic which is neglected part in psychiatry. Next, i need to tell you some concerns related to the manuscript .

1. There is no concestance ammong long title and short title. Some time you say the impacts and at the some time you used the effects. Because, the two words have some differences.

2. You did not stated how did you calculated the sample size and it is not clear. Samplig sizee is the main issue in scientific study.

3. You didnot clarify the Allocation ratio among control and intervention group?

4. The sampling techniques didnot stated?

5. The rationale of these study is not deeply mentioned or its shallow

6. You used DSM-IV text revision for diagnosis annd as evaluation criteria, whiich is not currently in use. Since 2017, DSM-5 is in use and why you didnot consider DSM-5 ?

7. You put none under acknowledge part. Is there any reason why you did so?

8. Please review all body part of these manuscript and check gramatic error, and sysematicallly connect with each other.

Regards!

Reviewer #2: Hello. Thank you for the opportunity to read your manuscript.

I think the findings are somewhat expected but interesting in this population. It would have been more helpful to have more details about the duration and dosages of antidepressants as you have mentioned as well as more objective data about the sleep disturbances.

6. PLOS authors have the option to publish the peer review history of their article (what does this mean? ). If published, this will include your full peer review and any attached files.

**Do you want your identity to be public for this peer review?** For information about this choice, including consent withdrawal, please see our Privacy Policy .

Reviewer #1: No

Reviewer #2: No

---

## [Author Response · Author response to Decision Letter 1]

25 Feb 2025

Dear prof. Marano,

Thank you very much for the invitation to revise and resubmit our manuscript, entitled "The effect of systematic antidepressant treatments in the early stages on sleep and impulsivity in bipolar euthymic patients: A cross-sectional study" to the PLoS One (ref: PONE-D-24-50851). We appreciate all efforts you have made and would like to thank the reviewers for their comments, which are very detailed and constructive and helped us to significantly improve the quality of the manuscript.

In our response, we provided point-by-point responses to all comments and made all additions and changes to the manuscript using track changes. Furthermore, we have corrected the grammatical errors and refined the expression throughout the manuscript.

We hope that you will find this revised version of our manuscript suitable for publication in the journal of PLoS One.

Yours sincerely,

Also, on behalf of the coauthors,

Chenghao Yang

Point by point response to reviewers

Reviewer 1.

First of all, I would like to thank Authours for providing this best topic which is neglected part in psychiatry. Next, i need to tell you some concerns related to the manuscript.

1. There is no concestance ammong long title and short title. Some time you say the impacts and at the some time you used the effects. Because, the two words have some differences.

Response:

Thanks for reviewer’s careful reading and constructive feedback regarding the consistency in terminology. We have recognized that although these terms are quite related, there are nuancedly distinct in the context of the current topic. In this regard, we agree with the suggestion from the reviewer and choose the “effect” as the unified term because it emphasizes the basic and neutral relationship between antidepressant treatments and observed outcomes. Accordingly, we have adjusted the long and short titles in the title page.

2. You did not stated how did you calculated the sample size and it is not clear. Samplig sizee is the main issue in scientific study.

Response:

Thank the reviewer for pointing out this important issue. As reviewer suggested, sample size calculation is crucial for the feasibility and scientific validity of a research, as it ensures that the study is adequately powered to detect real effects, while avoiding unnecessary resource waste or over-sampling, thereby enhancing the accuracy and reliability of the research. The study protocol, including the section on sample size calculation, had been published previously. We have referred it to reference No.33 in the section of “setting and subjects”.

Actions: (page 5, line 119-120)

Please refer to the published study protocol for experimental details.

3. You didnot clarify the Allocation ratio among control and intervention group?

Response:

Thanks for reviewer’s comment. Since our research was a cross-sectional study in which subjects did not receive any interventions and were grouped according to whether they had received systematic antidepressant treatments in the early stages rather than being assigned at random. So, we would say that it is inappropriate to engage the principle of “allocation ratio” here.

4. The sampling techniques didnot stated?

Response:

Thank you for raising this important issue. We recruited patients with bipolar disorder and grouped them according to whether they had received systematic treatment with antidepressants in the early stages. This grouping method is not random; instead, it deliberately selects patients with specific characteristics for the study, which is the purposive sampling. As described in the “Procedures and clinical assessment” section, the researchers involved in screening and allocation were different from those responsible for administering scale rating and data analysis, in order to decrease subjective bias. We have mentioned this and made a subtle revision in the “Setting and subjects” section. Please let us know if further information is required.

Actions: (page 4, line 99-102; page 5, line 109-110)

Based on a history of systematic antidepressant treatments in the early stages, patients were consecutively divided into the systematic antidepressant treatment (AT) group and the no systematic antidepressant (NT) group.

They had been in remission for at least 4 weeks prior to engage in the study.

5. The rationale of these study is not deeply mentioned or its shallow

Response:

We appreciate the reviewer for this helpful advice. If I am not mistaken, you are referring to the background of this study not being adequately explained. The primary goal was to explore the potential associations between antidepressant use, sleep and impulsivity of bipolar euthymic patients. The relationships between any of two variables are indirect or ambiguous. In this regard, it is valuable to explore it in view of the significant delay in diagnosis of bipolar disorder and widespread use of antidepressants before being properly diagnosed. Objectively speaking, the current topic has been overlooked, with all the focus placed on how to diagnose bipolar disorder early. However, the results from relevant studies are far from satisfactory. Therefore, it is important to address the effect of delayed diagnosis of bipolar disorder and inappropriate antidepressant treatment, which is the motivation and starting point for our research.

6. You used DSM-IV text revision for diagnosis annd as evaluation criteria, whiich is not currently in use. Since 2017, DSM-5 is in use and why you didnot consider DSM-5 ?

Response:

Thanks for reviewer’s valuable suggestion. The current study was conceptualized in 2019 and established in the mid-2020, by which time the DSM-5 and SCID-5 had not yet been adopted in China. Furthermore, the diagnostic criteria for bipolar disorder have not significantly changed between the DSM-IV-TR and DSM-5, except for the reclassification of the mixed episode as mixed features. We would say that the engagement of DSM-IV-TR in the study did not make significant bias for recruiting bipolar patients.

7. You put none under acknowledge part. Is there any reason why you did so?

Response:

We appreciate the reviewer's attention to acknowledgment section. We have revised it.

Actions: (page15, line 332-333)

We would like to express our gratitude to all the participants for their support and dedication to this study.

8. Please review all body part of these manuscript and check gramatic error, and sysematicallly connect with each other.

Response:

We thank the reviewer for these valuable suggestions. We have corrected the grammatical errors and refined the English writing throughout the manuscript. We have not highlighted every minor revision in the manuscript.

Reviewer 2.

Hello. Thank you for the opportunity to read your manuscript.

I think the findings are somewhat expected but interesting in this population. It would have been more helpful to have more details about the duration and dosages of antidepressants as you have mentioned as well as more objective data about the sleep disturbances.

Thank the reviewer for this insightful suggestion. We agree with the reviewer’s opinion and have illustrated it in detail in section of “limitations”. First, detailed information would improve the reliability of the research findings and support the broader dissemination and application of the results. Our study is a cross-sectional design that retrospectively collects medication data, including information from several years ago, which presents challenges in accurately obtaining all relevant details. We are concerned that inconsistencies in the level of detail could introduce bias. As a result, we have excluded such data from the analysis and discussion. Second, as you suggested, the objective data on the sleep, such as polysomnographic recording, would strengthen the reliability of the research results. But it is not feasible in terms of our experimental funding and conditions. Moreover, PSQI is commonly used in assessing sleep quality.

Actions: (page15, line 317-319)

Future research should incorporate longitudinal observations, utilize objective indicators, and minimize confounding factors to enhance the study’s validity.

---

## [Decision Letter · Decision Letter 1]

19 Mar 2025

The effect of systematic antidepressant treatments in the early stages on sleep and impulsivity in bipolar euthymic patients: A cross-sectional study

PONE-D-24-50851R1

Dear Dr. Yang,

We’re pleased to inform you that your manuscript has been judged scientifically suitable for publication and will be formally accepted for publication once it meets all outstanding technical requirements.

Kind regards,

Giuseppe Marano

Academic Editor

PLOS ONE

Additional Editor Comments (optional):

Reviewers' comments:

Reviewer's Responses to Questions

**Comments to the Author**

1. If the authors have adequately addressed your comments raised in a previous round of review and you feel that this manuscript is now acceptable for publication, you may indicate that here to bypass the “Comments to the Author” section, enter your conflict of interest statement in the “Confidential to Editor” section, and submit your "Accept" recommendation.

Reviewer #1: All comments have been addressed

2. Is the manuscript technically sound, and do the data support the conclusions?

Reviewer #1: Yes

3. Has the statistical analysis been performed appropriately and rigorously? 

Reviewer #1: Yes

4. Have the authors made all data underlying the findings in their manuscript fully available?

Reviewer #1: Yes

5. Is the manuscript presented in an intelligible fashion and written in standard English?

Reviewer #1: Yes

6. Review Comments to the Author

Reviewer #1: You came with nice topic and you did well. No more comments for you. All comments were incorporated

Regards!

7. PLOS authors have the option to publish the peer review history of their article (what does this mean? ). If published, this will include your full peer review and any attached files.

**Do you want your identity to be public for this peer review?** For information about this choice, including consent withdrawal, please see our Privacy Policy .

Reviewer #1: No

---

## [Editor Report · Acceptance letter]

PONE-D-24-50851R1

PLOS ONE

Dear Dr. Yang,

I'm pleased to inform you that your manuscript has been deemed suitable for publication in PLOS ONE. Congratulations! Your manuscript is now being handed over to our production team.

Kind regards,

on behalf of

Dr. Giuseppe Marano

Academic Editor

PLOS ONE